# Role of the MicroRNAs in the Pathogenic Mechanism of Painful Symptoms in Long COVID: Systematic Review

**DOI:** 10.3390/ijms24043574

**Published:** 2023-02-10

**Authors:** Samuel Reyes-Long, Jose Luis Cortés-Altamirano, Cindy Bandala, Karina Avendaño-Ortiz, Herlinda Bonilla-Jaime, Antonio Bueno-Nava, Alberto Ávila-Luna, Pedro Sánchez-Aparicio, Denise Clavijo-Cornejo, Ana Lilia Dotor-LLerena, Elizabeth Cabrera-Ruiz, Alfonso Alfaro-Rodríguez

**Affiliations:** 1Basic Neurosciences, Instituto Nacional de Rehabilitación LGII, Mexico City 14389, Mexico; 2Research Department, Universidad Estatal del Valle de Ecatepec, Ecatepec de Morelos 55210, Mexico; 3Escuela Superior de Medicina, Instituto Politécnico Nacional, Mexico City 11340, Mexico; 4Reproductive Biology Department, Universidad Autónoma Metropolitana, Mexico City 09340, Mexico; 5Pharmacology Department, Facultad de Medicina Veterinaria, Universidad Autónoma del Estado de México, Toluca 56900, Mexico; 6División de Reumatología, Instituto Nacional de Rehabilitación LGII, Mexico City 14389, Mexico; 7Neurociencias Clínicas, Instituto Nacional de Rehabilitación LGII, Mexico City 14389, Mexico

**Keywords:** microRNAs, long COVID, fatigue, pain, blood–nerve barrier, IL-6/STAT3

## Abstract

The ongoing pandemic of COVID-19 has caused more than 6.7 million tragic deaths, plus, a large percentage of people who survived it present a myriad of chronic symptoms that last for at least 6 months; this has been named as long COVID. Some of the most prevalent are painful symptoms like headache, joint pain, migraine, neuropathic-like pain, fatigue and myalgia. MicroRNAs are small non-coding RNAs that regulate genes, and their involvement in several pathologies has been extensively shown. A deregulation of miRNAs has been observed in patients with COVID-19. The objective of the present systematic review was to show the prevalence of chronic pain-like symptoms of patients with long COVID and based on the expression of miRNAs in patients with COVID-19, and to present a proposal on how they may be involved in the pathogenic mechanisms of chronic pain-like symptoms. A systematic review was carried out in online databases for original articles published between March 2020 to April 2022; the systematic review followed the PRISMA guidelines, and it was registered in PROSPERO with registration number CRD42022318992. A total of 22 articles were included for the evaluation of miRNAs and 20 regarding long COVID; the overall prevalence of pain-like symptoms was around 10 to 87%, plus, the miRNAs that were commonly up and downregulated were miR-21-5p, miR-29a,b,c-3p miR-92a,b-3p, miR-92b-5p, miR-126-3p, miR-150-5p, miR-155-5p, miR-200a, c-3p, miR-320a,b,c,d,e-3p, and miR-451a. The molecular pathways that we hypothesized to be modulated by these miRNAs are the IL-6/STAT3 proinflammatory axis and the compromise of the blood–nerve barrier; these two mechanisms could be associated with the prevalence of fatigue and chronic pain in the long COVID population, plus they could be novel pharmacological targets in order to reduce and prevent these symptoms.

## 1. Introduction

Coronavirus disease-19 (COVID-19) has spread to virtually every country in the world, infecting at the date of writing this review a total of 663,640,386 people, with an unfortunate number of 6,713,093 deaths (WHO); however, the number of deaths would have been a lot greater if not for the efforts of the medical and scientific community. Evidence of this is the outstanding number of clinical trials that were carried out in order to evaluate possible treatments to reduce the risk of developing severe disease or to prevent death; this has been already extensively reviewed elsewhere [1]. Another result of the efforts is the record-breaking time of the development and application of vaccines against severe acute respiratory syndrome coronavirus 2 (SARS-CoV-2), a great number of countries and companies teamed up in order to achieve vaccination in every country [2]. However, some obstacles such as the resistance of the population receiving these vaccines, as well as a partially biased distribution that has affected mainly developing and poor countries [3], has paved the way for the appearance of new and dangerous variants of the virus, which in turn provokes an increase in people with severe and moderate COVID-19.

At the beginning of the ongoing pandemic, no one could have foreseen that COVID-19 could cause long-lasting symptoms; however, it is now clear that this is a reality [4]. It is now known as long COVID and is characterized by the appearance or continuity of signs or symptoms derived from COVID-19 for more that 3 to 6 months [5]. The prevalence and incidence of long COVID has been extensively and thoroughly analyzed previously [6,7,8]. There is a myriad of symptoms comprehending long COVID ranging from hair loss, breathlessness, chronic coughing [7], to symptoms characteristic of the central [9] (CNS) and peripheral nervous systems (PNS) such as headache [10], memory impairment [11], attention deficit [12], generalized chronic pain [10] and fatigue [13]. These last ones have been attributed to an invasion if the virus to the nervous system [14], and two hypotheses of how this may happen exist: retrograde axonal transport and invasion of infected leukocytes through the blood–brain barrier [14,15,16], however, controversy still remains around this subject.

MicroRNAs (miRNAs, miRs) are small non-coding RNAs conformed by 17 to 22 nucleotides and are present in every mammalian cell; they modulate gene expression by silencing messenger RNA (mRNA). The miRNA seed region interacts with the untranslated region -3′ of the mRNA which leads to translation repression and mRNA degradation [17]. Since their discovery, the participation of miRNAs in pathophysiologic processes of several diseases has been described [18], thus, they are proposed as biomarkers for disease severity [19], for pharmacological treatment response [20] and as predictors of disease [21]. Based on the above, the objective of the present review was to show the prevalence of chronic pain-like symptoms of patients with long COVID and to propose miRNAs as important components in their development, based on miRs expression profiles in patients with acute COVID-19.

## 2. Material and Methods

Search strategies were developed for each database consulted: PubMed, Web of Science, litCOVID and Embase. Briefly, the search strategy was carried out using the indexation of MeSH (Medical Subject Headings) terms, combined with Boolean logical operators. The keywords and search terms employed were: (COVID-19) OR (long-COVID) OR (COVID-19 sequelae) OR (long-COVID syndrome) OR (long term COVID-19) OR (long term effects COVID-19) OR (post-acute COVID-19 syndrome) OR (post-COVID-19) AND (microRNA) OR and AND (miRNA). Articles published from March 2020 to April 2022 were included in the review. The reference list of the of articles included was checked in order to include any relevant article that was not found in the databases. The systematic review followed the Preferred Reporting Items for Systematic Reviewers and Meta-analysis (PRISMA) guidelines (Figure 1), plus it was registered in PROSPERO database with registration number CRD42022318992.

The inclusion criteria were as follows: (a) original articles written in English and Spanish were included, (b) articles that analyzed expression profiles of miRNAs by means of sequencing, microarrays or RT-qPCR only on humans, (c) articles that included patients with symptoms assessed 2 weeks or more after initial symptoms. The exclusion criteria were: (a) letters, editorials, commentaries, and case reports were omitted, (b) articles written in other language due to a lack of robust and accurate translation. The systematic and comprehensive search of the literature was conducted independently by two researchers R-L S and A-O K to evaluate each abstract and whole article; when discrepancies occurred, the full-text article was discussed among all the authors. The data that were extracted from each article were as follows: the methodology that was used to obtain the expression profiles of microRNAs, if available, the genes targeted by each miRNA, the source of the biological sample, the study design (in vivo, in vitro, in silico), the number of subjects, stage of the disease (COVID-19) of the patients, symptomatology of long COVID. The baseline characteristic is the presence of symptoms more than 4 weeks after the last day of acute infection by SARS-CoV-2. Data were collected in a spreadsheet elaborated with the agreement of all the authors.

## 3. Results

After performing the systematic review, a total of 22 articles that evaluated the expression profiles of microRNAs in COVID-19 patients and 20 articles regarding long COVID sequelae were selected. The PRISMA flow diagram that depicts the selection process is showed in Figure 1. Three studies were excluded because the symptoms evaluated were not of long COVID, another three studies did not employ a control group, five studies evaluated expression profiles of miRNAs from pathologies different of COVID-19 and one employed indirect evaluation of miRNA levels.

### 3.1. Long COVID Pain-Like Symtpomatology

It took some time to recognize and name long COVID, because early into the pandemic, the continuation of the symptoms was attributed to post-traumatic stress disorder caused by experiencing a novel disease with no known cure, or by the component of being infected and the consequential social isolation [22]. However, now it has been established that long COVID is an independent disease with a complex symptomatology [23], which comprises respiratory symptoms such as breathlessness [24], coughing, chest pain and dyspnea [25]; symptoms attributed to the CNS such as headache, generalized chronic pain, fatigue [26], attention deficit disorders [12], memory loss; even hair-loss, ageusia and hearing loss [27]. In the present review the focus will be on CNS symptoms, particularly painful manifestations, and fatigue (Table 1).

Given the novelty of the disease, there is still much to learn about its physiopathology in order to propose effective treatments and accurate diagnosis. However, some hypotheses have been proposed in order to shed light on this issue: a persistent inflammatory response, immune dysregulation, autoimmunity and viral persistence in several tissues, particularly in the CNS [28].

**Table 1 ijms-24-03574-t001:** Painful symptomatology of patients with long COVID.

Cite	Study Design	COVID-19 Severity (n)	Country/Nationality; Age (Mean)	Long COVID Pain-Related Symptoms	Long COVID Duration
[29]	Cross sectional	Mild (27), moderate (65), severe (18).	UK; 47, 57,62.	Fatigue 39%, myalgia 22%, Chest pain 13%	12 weeks
[30]	Cross sectional	Noninvasive ventilation (21), invasive ventilation (7)	Italy; 56.5	Fatigue 53.1%, arthralgia 27.3%, chest pain 21.7%	60.3 days
[31]	Prospective	Moderate (101), severe (29)	France	Arthralgia 21%, chest pain 17%	60 days
[32]	Cross sectional	488	USA; 62	Unable to return to normal activity 38.52%, New or worsening difficulty completing activities of daily living 11.88%	60 days
[33]	Cross sectional	307	China	Soreness in the throat 17.9%, Fatigue 11.4% myalgia and arthralgia 8.8%	
[34]	Cross sectional	Non-Hospitalized (2133), Visited ER or Urgent Care (1312), Hospitalized (317)	White (85.3%), Hispanic, Latino, Spanish Origin (3.7%), Asian, South Asian, SE Asian (3.3%), Black (2%), Middle Eastern, North African (1.7%), Indigenous Peoples (1.6%), Pacific Islander (0.1%), Other (2.5%)	Headache 77%	7 months
				Fatigue 80%, Headache 53.60%	6 months
[35]	Cross sectional	Neurological complications (196), Controls (186)	White (44%, 41%), Black (11%, 14%), Asian (10%, 4%), Native American/Pacific Islander (0.5%, 1%), American Indian (0.5%, 1%); 68 and 69	Difficulty completing activities of daily living	6.7 months
[36]	Cross sectional	Neurological complications (113), non-neurological complications (129)	White (43%, 34%), Black (8%, 12%), Asian (4%, 4%), American Indian/Alaska Native (0, 1%), Other (19%, 16%), Prefer not to answer (26%, 33%); 64 and 65	mRS > 0 75%, Barthel index < 100 64%, T-MoCA < 18 50%, Fatigue 9%	12 months
		Neurological complications (86), non-neurological complications (88)		A 6 months,88% had at least one abnormal metric. A 12 months, 84% had at least one abnormal metric.	6 and 12 months
[37]	NA	Non-Hospitalized (2001), Hospitalized (317)	Netherlands and Belgium; 47	Fatigue 87%, Chest tightness 44%, Headache 38%, Muscle pain 36%, Pain between shoulder blades 33%, Sore throat 26%	79 days
[38]	Cross sectional	Ward patients (68), UCI patients (32)	White (79.4%, 59.4%), Mixed (1.5%, 0), Asian or Asian British (2.9%, 25%), Black or Black British (7.4%, 9.4%); 70.5 and 58.5	Worsened pain/discomfort 14.7% and 28.1%	14 ± 10.3 días
[39]	Cross sectional	Severity scale: 3 (439) 4 (1172) 5-6 (122)	China; 57	Fatigue 81.20%, Joint pain 14.53%, Chest pain 8.55%, Sore throat 4.27%, Myalgia 3.42%, Headache 2.56%	205 days
[24]	Cross sectional	Oxygen alone (217), ICU (54), Intubation (47)	British Caucasian (38.8%), Other Caucasian (17.1%), British Asian (6.5%), Other Asian (10.3%), Black British (6.8%), Other black (7.6%), Other ethnicity (13.9%); 59.9	Fatigue 67.3%, 73.3%, 76.9%	54 days
[40]	Cross sectional	No pneumonia (20), Mild (15), Severe pneumonia (106)	Spain; 62	Fatigue 68.1%, Myalgia and arthralgia 38.3%, Headache 34.8%, Severe headache 18.4%	77 days
[41]	Cross sectional	Symptoms at acute phase but not at follow-up (178)	Faroe Islands	Headache 56.7%, Fatigue 48.9%	81 days
[42]	Cross sectional	Mild (57),	Italy, 62.3	Fatigue 29.8% Myalgia 24.7% Headache12.5%	6 months
		Moderate (77)	Italy, 67.3	Fatigue 31.2% Myalgia 31.6% Headache 6.5%	
		Severe (31)	Italy, 63.2	Fatigue 48.4% Myalgia 38.7% Headache 12.9%	
[43]	Cross sectional	Mild (16), Severe (4)	Germany	Fatigue 55%, Myalgia 15%, Headache 10%,	225.3 days
[44]	Cross sectional	Normal HADS-A/D (70), Pathological HADS-A/D (30)	Italy; 55 and 56	Pain 7.10%, 20%	46 days
[45]	Cross sectional	128	Ireland; 49.3 ± 14.3 and 49.7± 16	Fatigued 52.3%	8–12 weeks
[46]	Cross sectional	Non-Hospitalized (79), Hospitalized (55), ICU (19)	White (70.9%, 81.8%, 73.7%), Asian (21.5%, 10.9%, 15.8%), Hispanic (2.5%, 0, 0), African (5.1%, 7.3%, 10.5%); 40.2, 56.4 and 54.5	Fatigue 48%	75 days
[47]	Cross sectional	Anosognosia (26)	Switzerland; 56.58 and 56.49	Physical pain 85.19%, 69.84%	227.07 ± 42.69 days
		Nosognosia (76)		Sore throat 0, 1.3%, Muscle pain 7.7%, 10.5%, Fatigue 23.1%, 53.9%, Chest pain 0, 2.6%, Headache 7.7%, 13.2%	6–9 months

### 3.2. MicroRNAs and COVID-19

The dysregulation of miRNAs in several diseases is well-known and it has been explored as a novel strategy for diagnostics and therapeutics. In light of the recent pandemic, a great number of studies were published in which expression profiles of COVID-19 patients were evaluated; in the current review we sought to identify which of the miRNAs that were abundantly altered in the acute phase of the disease, and could putatively be involved in long COVID based on the mechanisms and genes regulated by them. By means of the systematic review, eighteen miRNAs (Table 2) were identified to be commonly deregulated in at least three of the studies included: miR-21-5p, miR-29a-3p, miR-29b-3p, miR-29c-3p, miR-92a-3p, miR-92b-3p, miR-92b-5p, miR-126-3p, miR-150-5p, miR-155-5p, miR-200a-3p, miR-200c-3p, miR-320a-3p, miR-320b, miR-320c, miR-320d, miR-320e, miR-451a.

#### 3.2.1. miR-21-5p

The miR-21-5p has been extensively associated with immunological processes, and it has been categorized as an “immuno-miR” [70], because among other processes, it is important for maintaining the effector phase of T-cells. In the studies included in the present review, the expression profile data of miR-21-5p differed. Garg et al. [53] performed two different evaluations: in the first one, named discovery cohort, they compared the serum of critically ill patients with COVID-19 vs. healthy controls, and they found an upregulation of miR-21-5p in the COVID-19 group. However, in the validation cohort they compared serum of mechanically ventilated patients with COVID-19, invasively ventilated Influenza-ARDS patients and healthy controls, finding no significant difference between the COVID-19 and healthy control group. Saulle et al. [67] evaluated the peripheral blood mononuclear cells (PBMCs), plasma and placenta of pregnant women infected and uninfected with SARS-SoV-2; they also found an upregulation of miR-21-5p but only in plasma, and the authors proposed that this upregulation resembles a defensive attempt of the organism to interfere with viral replication by directly targeting it gene expression, based on a possible correlation between miR-21-5p and four predicted binding sites with SARS-CoV-2. Eichmeier et al. [52] also found an upregulation of miR-21-5p in nasopharynx tissue of a SARS-CoV-2 positive group when comparing to a negative group. 

In contrast, Sabbatinelli, et al. [66] in a sample of 29 COVID-19 patients under treatment of Tocilizumab, a monoclonal antibody against interleukin-6 (IL-6) receptor, reported a significant decrease in miR-21-5p expression levels between patients and healthy controls, plus, a tendency for upregulation was observed when comparing basal time and 72 h after TCZ administration; notably, a significant correlation was found between miR-21-5p and IL-6 and D-dimer levels. At the same line, a thorough study performed by Tang et al. [71], in which they analyzed severe and moderate patients of COVID-19, showed a significant downregulation of miR-21-5p in red blood cell-depleted whole blood samples. Plus, the authors propose miR-21-5p as a candidate biomarker that may contribute to COVID-19 pathogenesis and serve as therapeutic targets, which then they validated with single-cell RNA seq data from two previously published papers. Lastly, Keikha et al. [59] performed a study in which they segregated COVID-19 patients into six groups according to the severity of the disease: critical, moderate, mild, asymptomatic and a healthy control group. They analyzed whole blood small RNA and found a significant decrease in miR-21-5p expression only in the critical illness group and a negative significant correlation between this miRNA and its target mRNA IL-12p53 at all severity grades.

The difference between the data of the displayed studies could be explained by the heterogeneity of the samples employed; two studies coincide in the evaluation of serum from patients, but the others evaluated plasma, RBCD whole blood, and whole blood. It is known that miR-21-5p plays a key role on the immune system [70]. Li et al. [72] mentioned that this miRNA is an important regulator in the infection from hepatitis C and B virus and Epstein-Barr virus; the authors found that miR-21-5p is important in maintaining the effector phase of Treg cells and that it is highly involved in the modulation of interferons (IFN), nuclear factor-κB (NF-κB) and signal transducer and activator of transcription (STAT3). In line with this, in a recent study, De Melo et al. [73] found that the miR-21-5p PGE2/IL-10 axis controls the inflammatory profile of macrophages, which is critical for sepsis outcomes, plus they found that it is highly expressed in macrophages and neutrophils after sepsis. Another explanation for the discrepancies could be to the complex nature of the regulation exerted by miR-21-5p; some authors propose it as an anti-inflammatory and some as a pro-inflammatory regulator (31, 47, 30 de De Melo, 2021).

Another important role of miR-21-5p is its participation in nociception; its differential expression has also been shown in several conditions. Hori et al. [74] evaluated the neuropathic pain model of sciatic nerve ligation and found that 26 miRNAs were upregulated and 4 were downregulated; among them, miR-21-5p was significantly and drastically upregulated in the dorsal foot ganglion (DRG) from 3 to 7 days after nerve ligation. Plus, Leinders et al. [75] evaluated a cohort of patients that suffered from neuropathic pain from different etiologies; in them the authors found that miR-21-5p expression is significantly higher in white blood cells and the sural verve of patients with polyneuropathies. In a more recent study, Reinhold et al. [76] found that patients with complex regional pain syndrome (CRPS), as well as a rodent model of neuropathic pain, presented a significant higher expression of miR-21-5p; the authors concluded that this miRNA is a major switch in neuropathic pain due to its role in affecting pain behavior through different pathways, including inflammation-independent barrier impairment.

#### 3.2.2. miR-29a-3p and miR-29b-3p

Donyavi et al. [51], evaluated the expression levels of miR-29a-3p in 18 COVID-19 infected patients, and in 15 healthy controls; the researchers isolated the PBMC and found that miR-29a-3p was overexpressed between these two groups. Interestingly, they also found that in patients in the recovery period of COVID-19, this expression was higher when comparing with the acute phase. Eichmeier et al. [52] identified, by means of comparing miRNA sequences with the total abundance obtained by small RNA seq, that miR-29-a-3p expression was upregulated in nasopharyngeal samples of SARS-CoV-2-positive patients; however, when validating with RT-qPCR, no significant difference was found. Along with this, Saulle et al. [67] found an upregulation of miR-29a-3p in plasma of SARS-CoV-2-infected pregnant women. In contrast, miR-29a-3p expression was evaluated in a longitudinal study of COVID-19 patients with severity grades I through V and healthy controls; the authors found that it was significantly downregulated in severity grade groups IV and V in comparison with other severities and healthy controls. Next, after two weeks, the researchers segregated patients that responded to treatment and those who did not and found that in the former, miR-29a-3p expression continued to be significantly downregulated, contrasting with the respondent group where the expression significantly rose [60]. In line with this, Greghl et al. [56] evaluated the plasma of a heterogeneous sample of COVID-19 patients; the authors report that miR-29a-3p and -29b-3p were downregulated. Centa et al. [50] performed an analysis of formalin-fixed paraffin-embedded post mortem lung biopsies of COVID-19 patients and found that miR-29b-3p was downregulated when comparing to a control group. Finally, in an analysis of peripheral blood of moderate and severe patients with COVID-19, Liu et al. [61] found that the miR-29-b-3p was significantly downregulated in severe patients when compared to the ones in the moderate group.

#### 3.2.3. miR-92a-3p, -92b-3p and -92b-5p

Gonzalo-Calvo et al. [55] performed a study that included 79 COVID-19 patients, segregated into two groups, with and without intensive care unit (ICU) admission. The authors found a significant decrease of miR-92a-3p in ICU patients, plus they identified two significant correlations, the first one between the mortality of ICU patients and miR-92a-3p expression levels, and the second one being an inverse correlation between plasma levels of mir-92a-3p and the total number of days in ICU stay. In contrast, Fayyad-Kazan et al. [58] evaluated the miRNA expression profiles of 33 COVID-19 patients and 10 healthy controls and found that in plasma, miR-92a-3p was significantly upregulated. This contrast may be explained by two factors: the heterogeneity in the COVID-19 population in Kazan’s study (they included patients with different severity degrees) and the fact that Gonzalez-Calvo did not employ a healthy control group. Wu et al. [69] evaluated nasopharyngeal swabs of seven SARS-CoV-2-positive and six SARS-CoV-2-negative samples, and they found a downregulation of miR-92b-3p and an upregulation of miR-92b-5p. On the other hand, Parray et al. [64] found a downregulation of miR-92b-5p, however, they only compared severe vs. mild cases.

#### 3.2.4. miR-126-3p

Across all studies included in the present review, miR-126-3p expression was found to be downregulated, Garg et al. [53] compared miRNAs expression profiles in the plasma of ICU patients with severe COVID-19 requiring invasive ventilation, mechanically ventilated Influenza-ARDS patients, and healthy controls; the authors found a significant decrease only in the discovery cohort. Grehl et al. (2021) also found a significant downregulation in severe when compared with mild cases. An interesting result comes from the study of Keikha et al. [60]; apart from finding a downregulation of miR-126-3p, they evaluated the patients who responded to treatment and those who did not. In the former, miR-126-3p expression returned to normal levels at week 2 and in the latter the downregulation was greater at 2 weeks. In line with this, Nicoletti et al. [63] compared miRNA expression profiles of patients with COVID-19 and healthy controls, and they found a significant downregulation of miR-126-3p, but between severe and mild patients, this downregulation was not present, perhaps to the lower sample size between these two groups. Finally, Sabbatinelli et al. [66] analyzed serum of COVID-19 patients and found a downregulation of miR-126-3p; they also found a significant positive correlation with miR-126-3p and neutrophils levels, and a significant negative correlation with IL-6 and D-dimer.

#### 3.2.5. miR-150-5p

Wilson et al. [68] evaluated plasma of hospitalized COVID-19 patients and segregated them according to the severity of the disease; they found that mild cases presented a significant upregulation of miR-150-5p, and they also found an upregulation of IL-6 in severe cases when compared with mild and moderate ones. Saulle et al. [67] also found an upregulation in plasma of pregnant women infected with SARS-CoV-2. In contrast, Gonzalo-Calvo et al. [55] found decreased levels of miR-150-5p when comparing plasma of ward vs. ICU patients with COVID-19. In line with this, Akula et al. [49] found a downregulation of miR-150-5p expression in COVID-19 patients with moderate to severe disease, and the researchers also found that in vitro miR-150-5p mimics had the ability to lower SARS-CoV-2 infection in cells by targeting the *nsp10* gene. Finally, in severe/critical vs. mild COVID-19 patients, Nicoletti et al. [63] found a significant downregulation of miR-150-5p expression. Thus, the disparity in the results above shown may be due to the severity state of the patients sample; in severe to critical stages of the disease miR-150-5p appears to be downregulated, which has been seen in other disease states [77], and in mild cases miR-150-5p expression can be upregulated as an early protective mechanism.

#### 3.2.6. miR-155-5p

Another miRNA that is greatly involved in several nervous and non-nervous processes is the miR-155-5p; this miRNA was found to be commonly upregulated in all but one of the studies included in the present review. Mc Donald et al. [78] found that miR-155-5p was downregulated in COVID-19-positive nasopharyngeal swabs. However, an upregulation of this miRNA has been linked to several disease states, mainly linked to inflammation [79]. The upregulation found by Keikha et al. [60] in serum samples of critical COVID-19 cases, plus the negative correlation with its target SOCS1, shows the role of miR-155-5p in promoting neuronal inflammation. Furthermore, Haroun et al. [57] found a significant upregulation of miR-155-5p in severe COVID-19 patients, and they observed the same upregulation in non-survivors when compared to survivors, prompting a significant positive correlation with mortality. The same upregulation was found by Garg et al. [53], Donyavi, et al. [51] and Saulle et al. [67].

#### 3.2.7. miR-200a-3p and miR-200c-3p

The mir-200 family, particularly, miR-200a-3p and miR-200c-3p, were found to be deregulated in the studies included in the present review. Both are implicated in proinflammatory and oxidative signaling; miR-200c-3p expression is induced by NF-κB, which, in turn, leads to ROS increases and NO decreases [80]. Plus, there has been reports of miR-200c-3p targeting ACE2, leading to a decrease in the protein [81]. Abdolahi et al. [48] found, in the peripheral blood of 30 COVID-19 patients, a significant downregulation of miR-200c-3p when compared with 18 healthy controls at admission time, however, at discharge time this expression increased in the COVID-19 group. The researchers also found a significantly negative correlation between IL-6 expression in serum and miR-200c-3p; IL-6 was significantly upregulated. In an extensive study, Zheng et al. [82] analyzed PBMC of 18 COVID-19 patients, of which 6 coursed the disease with mild, 7 with moderate and 5 with severe symptoms; among other things, the authors found that the regulation of T-cell differentiation by differentially expressed miRNAs occurred at the rehabilitation stage and among the downregulated ones was miR-200c-3p. In contrast, Pimenta et al. [65] reported a significant upregulation of miR-200c-3p when evaluating saliva samples of 111 patients with COVID-19; the authors segregated the patients into four groups according to disease severity, group I were negative COVID-19 patients, II were non-hospitalized patients, III were hospitalized but not ICU patients and finally IV were patients that were admitted to the ICU. The authors observed a strong pattern according to severity; the more severe the patients were, the higher the expression of miR-200c-3p. The discordance found in the present studies could be due to the different samples employed by the researchers; saliva could present more biases than peripheral blood or serum.

#### 3.2.8. miR-320a, miR-320b, miR-320c, miR-320d and miR-320e

Interestingly, all members of the extensive miR-320 family were commonly upregulated in all the studies included in the present review. Fayyad-Kazan et al. [58] evaluated the plasma of 33 COVID-19 patients and compared it with 10 healthy controls; they identified an upregulation of miR-320a. In plasma as well, Grehl et al. [56] found an upregulation of the entire subset of miR-320, miR-320a, b, c, and d, allowing a distinction between severe and mild cases. Next, nasopharyngeal swabs (NPS) of seven positive and six SARS-CoV-2-negative patients were collected by Wu et al. [69]; in them, they found, among many other interesting results, an upregulation of miR-320b and c. Four patients with severe, four with mild COVID-19 and four healthy controls were evaluated by Nicoletti et al. [63]; the authors found that miR-320b, c and d were upregulated in whole-blood samples. A similar categorization by severity was performed by Wilson et al. [68], and in their extensive miRNA/cytokine/chemokine integration, the first cohort evaluated was formed by 20 mild, 21 moderate and 17 severe COVID-19 patients; the authors reported a significant upregulation of miR-320e.

#### 3.2.9. miR-451a

Finally, Gonzalo-Calvo et al. [55] found, in nasopharyngeal swabs of COVID-19-hospitalized patients, a downregulation of miR-451a; plus, they found a signature of three miRNAs: miR-148a-3p, miR-486-5p and miR-451a, associated with ICU stay. Grehl et al. [56] in their study of plasma of mild and severe COVID-19 patients also found a downregulation of miR-451a. This was again found by Wilson et al. [68] when they analyzed 58 hospitalized COVID-19 patients; interestingly the authors also found in severe patients that IL-6, IL-10 CCL20 and miR-451a are key correlates with the fatality of the disease.

## 4. Discussion

In the following sections the mechanisms regulated by these miRNAs and how they relate to the chronic nociceptive process that is present in a significant proportion of long COVID patients, will be discussed.

### 4.1. Exacerbated and Persistent Inflammatory Response

Cytokines participate in a variety of physiological processes such as immunity [83], embryonic development [84], aging [85] and particularly inflammation; the release of proinflammatory cytokines in response to noxious stimuli such as mechanical, thermal or chemical injury and infection generates a normal response in which immune cells are recruited to the site in order to achieve homeostasis [86]. However, chronic inflammatory states could arise in consequence of a disease [87], and a clear example is the “cytokine storm” present in patients with acute COVID-19 [88].

One of the mechanisms that we identified to be commonly regulated by miRNAs expressed in COVID-19 patients is the interleukin-6/signal transducer and activator of transcription 3 (IL-6/STA3) axis, and this has been proposed previously [89]. This signaling pathway begins when the cytokine IL-6 interacts with its membrane-bound receptor (IL-6R), then the complex activates Janus kinase 1 and 2 (Jak1, Jak2), which in turns phosphorylate STAT3; there is a non-canonical pathway where transforming growth factor-β1 (TGF-β1) interacts with its receptor and Jak2 exerts phosphorylation on STAT3 [90]. STAT3 then translocates to the nucleus and the transcription of genes involved in cell differentiation, survival, proliferation, inflammation and immunity is activated [91]. Lastly, there is a negative feedback loop mediated by the suppressor of cytokine signaling 3 (SOCS3). Transforming growth factor-α (TNF-α) and interleukin-1 β also activate STAT3 but in an indirect manner [92]; the former enhances the production of IL-6 from non-immune cells in a manner dependent on nuclear factor-κB (NF-κB), prompting a positive-feedback amplification loop of the IL-6/STAT3 axis. Plus, the IL-6/STAT3 axis is heavily involved in immunity signaling; it inhibits T naïve cell differentiation into Tregs. However, when Tregs are present, IL-6/STAT3 promotes its differentiation into pro-inflammatory Th17 cells [93] (Figure 2). The IL-6/STAT3 axis is heavily involved in the pathogenic process of SARS-CoV-2 infection, and some researchers suggest that modulation of this pathway could be an important target for pharmacological therapy against the infection [94].

In the present review we identified that miR-21-5p was commonly downregulated in patients with active SARS-CoV-2 infection; there is a feedback regulation between the miR-21-5p and IL-6/STAT3 axis [73]. Thus if miR-21-5p is downregulated, the axis partially loses regulation, prompting an exacerbated inflammatory response. Furthermore, miR-29a-3p and -29b-3p depletion has shown to increase circulatory levels of IL-6 [95], the direct activator of the IL-6/STAT3 axis, and in patients with COVID-19 these two miRs were found to be downregulated. miR-155-5p was upregulated in five of the studies included in the present review and this has been associated as well with an increase of IL-6 and TNF-α [96]; plus, NF-κB activation, found in the IL-6/STAT3 axis, increases miR-155-5p expression [97], which results in an overactivation of the axis. Finally, miR-155-5p targets and decreases the suppressor of cytokine signaling 1 (SOCS-1) [97], which is also a regulator of cytokine signaling trough the JAK–STAT pathway [98]. In the same line, miR-200a-3p overexpression downregulates SOCS-6 [99], also a JAK–STAT regulator, and in COVID-19 patients this miRNA is in fact overexpressed; moreover, miR-200c-3p is positively correlated with IL-6 levels [100] and in one study included in the review this miRNA was overexpressed. An interesting finding is that across all studies, the miR-320 family (miR-320a-3p, -320b, -320c, -320d, -320e) was upregulated in COVID-19 patients, and a recent study found that this family is positively correlated with the IL-6/STAT3 axis, causing exacerbated cardiac fibrosis caused by an elevated synthesis of IL-6, p-STAT3 and TGF-β [101]. Finally, miR-451a, which is known for targeting and inhibiting the IL-6/STAT3 axis [102], was found to be downregulated in COVID-19 patients. With all of the above we can conclude that the deregulation of miRNAs caused an imbalance in the inflammatory response towards an hyperinflammatory state of the IL-6/STAT3 axis (Figure 2).

### 4.2. Compromise of the Blood–Nerve Barrier

The blood–nerve barrier (BNB) surrounds peripheral nerves and protects them from noxious stimuli; it is formed by three layers, the endothelium in endoneurial blood vessels, the perineurium surrounding nerve fascicles, and the autotypic junctions of Schwann cells [103]. An increase in BNB permeability has been shown in several diseases in which neuropathic pain is a common symptom [104].

Claudin 1 is the major known sealing tight junction protein, thus a key factor in BBB and BNB permeability; in the latter, is expressed in all three parts: in the perineurium, endoneurial vessels and autotypic junctions. Previous research has showed that a reduction in claudin-1 expression leads to the appearance of neuropathic pain symptoms in animal models [103] and some evidence suggest that this is the same in humans [104]. Interestingly, miR-155-5p directly targets and suppresses claudin-1 and in the present review an upregulation of miR-155-5p across five studies was present; this could suggest that the long-term neuropathic pain-like symptoms present in long COVID could be due to a suppression of claudin-1 and a subsequent increase in endothelial permeability [79]. Furthermore, an indirect modulation of claudin-1 in long COVID can also be hypothesized; by means of the modulation of matrix metalloprotease-9 (MMP-9), an important regulator of axon demyelination and extension [105]; MMPs are predominantly expressed in Schwann cells of peripheral nerves under normal conditions and are upregulated in peripheral nerve injury [106,107]. Several miRNAs of the ones here reviewed exert direct or indirect modulation of MMP-9; miR-21-5p has been shown to impair BNB integrity by inhibiting RECK, giving rise to MMP-9 and further downregulating claudin-1 [108]; plus, a direct induction of MMP-9 expression by miR-21-5p was also shown [76]. In the present review, the levels of miR-21-5p were divided between being down- and upregulated; the former could be associated with a mechanism of protection and the latter with pathogenic one. Furthermore, miR-200c-3p, which was upregulated in one of the studies included, is positively correlated with the expression of MMP-9 [100]. Finally, a recent report found that an administration of lentivirus expressing miR-451a led to an upregulation of said miRNA in malignant glioma cell lines, which caused a decrease in MMP-9, as confirmed by Western blot [109]. The downregulation found in the three studies reviewed, one in severe COVID-19, could indicate an increase of MMP-9 in these patients prompting long term consequences in the PNS, leading to the appearance of peripheral pain (Figure 3). However, a claudin-1-independent activity of MMP-9 can also be plausible; elevated MMP-9 was shown to modulate hyperalgesia by initiating IL-1β cleavage and microglial activation [110,111]. Plus, Schwann cells have been observed to release MMP-9 after mechanical damage to axons, initiating macrophage infiltration through the BNB and degradation of the myelin basic protein [105].

## 5. Conclusions

This review presented the mechanisms whereby miRNAs could be proposed as prognostic predictors and as therapeutic targets in patients with long COVID; now well identified as sequela of SARS-CoV-2 infection, particularly in painful manifestations and fatigue, symptoms that have been poorly studied and heavily reported by patients after COVID-19. In the present review, 18 miRNAs that were abundantly altered in the acute phase of COVID-19 were identified, divided into 9 families. These miRNAs regulate several important genes (Table 3).

The evidence displayed above paints a picture of the putative mechanisms that could be participating in the pathogenesis of the altered nociceptive response present in patients with long COVID, and said mechanisms are not isolated from one another, as is shown in the complex interplay of miRNAs summarized in the present review; the IL-6/STAT3 axis was found to be heavily involved in the pathogenic process of SARS-CoV-2 infection. We can conclude that the dysregulation of miRNAs caused an imbalance in the inflammatory response towards a hyperinflammatory state of the IL-6/STAT3 axis. Some studies suggest that modulating this pathway could be an important target for drug therapy against infection. Plus, we cannot fail to mention the importance of the BNB, which presents an increase in permeability in several pathologies in which neuropathic pain is a frequent symptom. The miRNA regulation here described could suggest that the symptoms similar to long-term neuropathic pain present in long COVID could be due to a suppression of claudin-1 and a subsequent increase in endothelial permeability, initiating macrophage infiltration through BNB and degradation of the myelin basic protein that leads to an increase in BNB permeability causing the appearance of pain symptoms.

We can conclude that miRNAs play a fundamental role as regulators of gene expression at the post-transcriptional level through RNA interference pathways, which could allow us to propose them as novel markers in long COVID, particularly in patients with symptoms such as pain or fatigue. Studies focused on analyzing the exosome epigenetic print in patients with long COVID, contemplating and segregating them according to the diversity of sequelae, could provide sound evidence in order to propose pharmacological and non-pharmacological therapies focused in alleviating symptoms and improving quality of life.

## Figures and Tables

**Figure 1 ijms-24-03574-f001:**
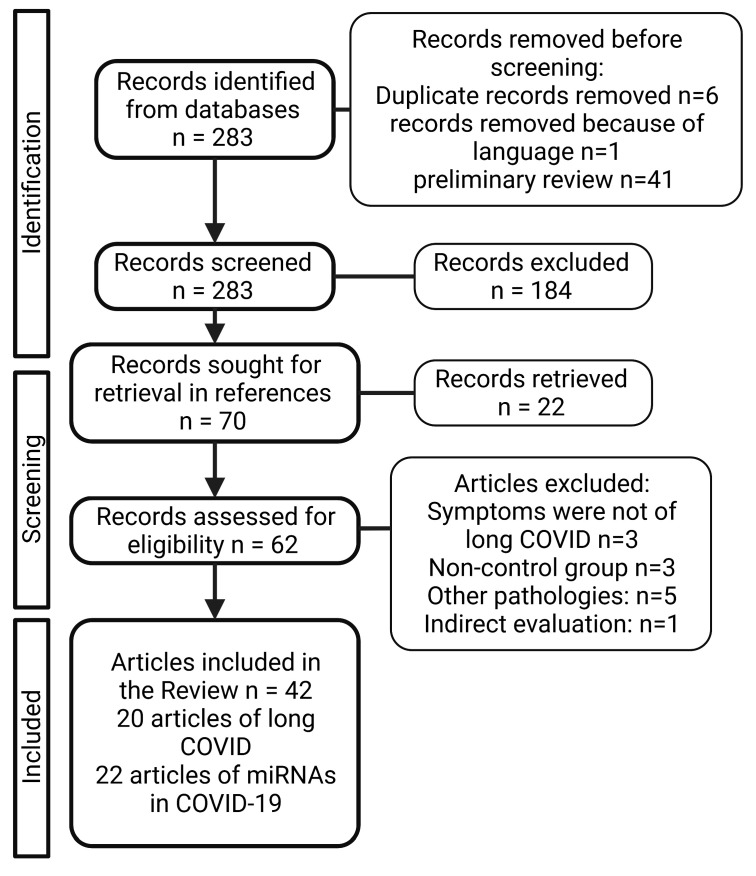
PRISMA flow diagram.

**Figure 2 ijms-24-03574-f002:**
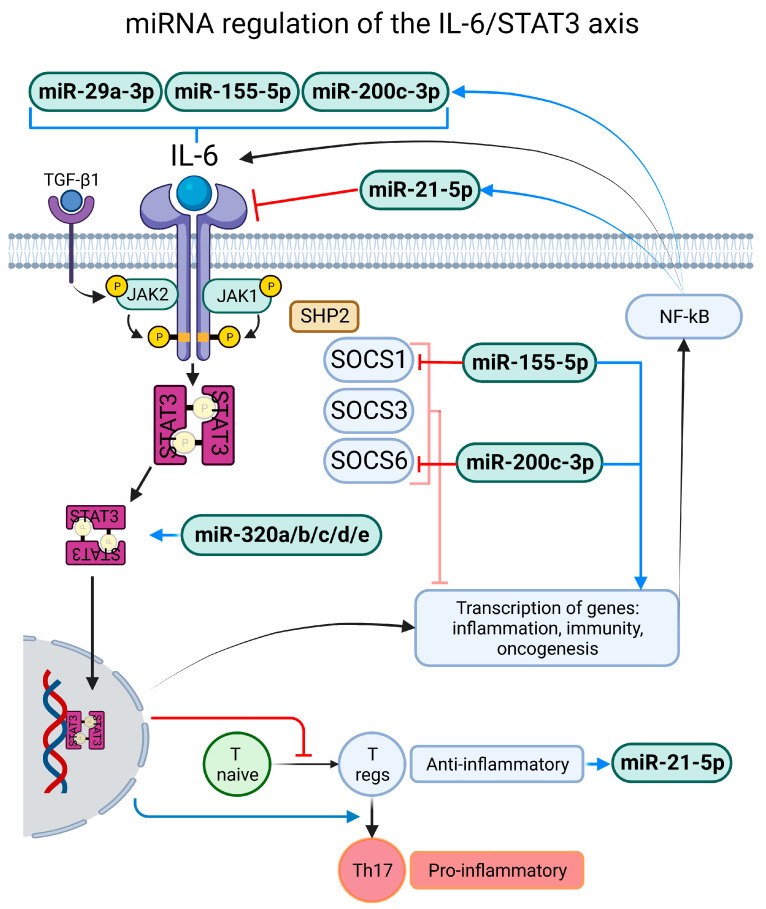
The IL-6/STAT3 axis and regulation exerted by miRNAs; the downregulation of miR-29a-3p, and the upregulation if miR-155-5p and miR-200c-3p have been associated with an increase in IL-6, which in turns acts on its membrane receptor causing the phosphorylation of STAT3. The IL-6R expression is regulated by miR-21-5p; plus, the miR-320 family has been shown to be associated with the activation of the IL-6/STAT3 axis. When STAT3 translocates to the nucleus it causes the expression of inflammation, immunity and oncogenesis genes and the inhibition of the differentiation of the T naïve cells into the anti-inflammatory T regs; instead, it caused the differentiation of existing T regs into the pro-inflammatory Th17 cells. The regulation of the axis is carried out by SOCS1, 3 and 6; however miR-155-5p downregulates SOCS1 and miR-200c-3p SOCS6, causing an overactivation and further exacerbated release of NF-κB which in turns upregulates miR-155-5p and miR-200c-3p.

**Figure 3 ijms-24-03574-f003:**
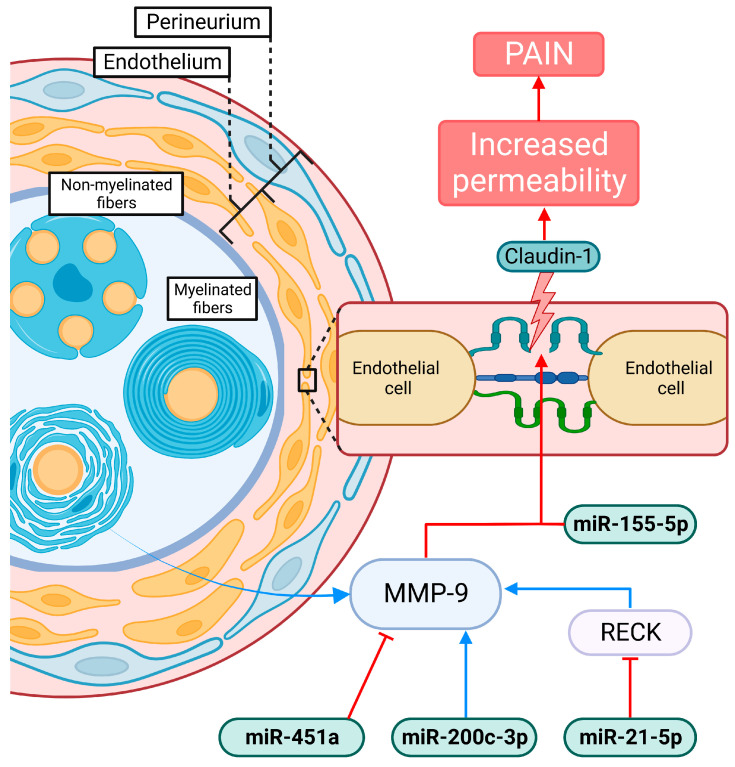
The blood–nerve barrier BNB compromise. The protein claudin-1 is the major sealing tight junction protein in the endoneurium; it is downregulated by MMP-9, which is upregulated in nerve damage. Plus, miR-200c-3p directly upregulates MMP-9; indirectly, by means of RECK inhibition, miR-21-5p also upregulates this matrix metalloprotease, and, in contrast, miR-451a downregulates it. Thus, the increase of MMP-9 leads to an increase in the BNB permeability which finally causes the apparition of pain symptoms.

**Table 2 ijms-24-03574-t002:** microRNA expression in COVID-19 patients.

Author	miRNAs	Level Changes	Country/ Nationality	N (Exp—Control)	Sample
Abdolahi et al. [48]	miR-200c-3p	↓	Iran	30—18	Peripheral blood
Akula et al. [49]	miR-150-5p	↓	USA	12—8	Plasma
Centa et al. [50]	miR-29b-3p	↓	Brazil	9—10	Lung tissue
Donyavi et al. [51]	miR-29-3p, 155-5p	↑,↑	Iran	18—5	PBMC
Eichmeier et al. [52]	miR-21-5p, 29a-3p, 200a	↑,↑,↑	NA	10—10	Nasopharyngeal tissue
Garg et al. [53]	miR-21-5p, 126-3p, 155-5p,	↑,↓,↑	Germany	18—15	Serum
Giuliani et al. [54]	miR-320b	↑	Italy	6—6	Serum
Gonzalo-Calvo et al. [55]	miR-92a-3p, 150-5p, 451a	↓,↓,↓	Spain	84—79	Plasma
Grehl et al. [56]	miR-29a-3p, 126a-3p, 320a-3p, 320b, 320c, 320d, 451a	↓,↓,↑,↑,↑,↑,↓	Caucasian	8—2	Plasma
Haroun et al. [57]	miR-155-5p	↑	Egypt	150—50	Peripheral blood
Fayyad-Kazan et al. [58]	miR-92a-3p, 320a	↑,↑	Lebanon	33—10	Plasma
Keikha et al. [59]	miR-21-5p, 155-5p	↓,↑	Iran	103—20	Serum
Keikha et al. [60]	miR-29a-3p, 126-3p	↓,↓	Iran	103—20	Serum
Liu et al. [61]	miR-29b-3p	↓	China	10—4	Peripheral blood
McDonald et al. [62]	miR-29b-3p, 155-5p	↓,↓	USA	10*	Serum
Nicoletti et al. [63]	miR-126-3p, 150-5p, 320b, 320c, 320d	↓,↓,↑,↑,↑	Brazil	8—4	Plasma
Parray et al. [64]	miR-92b-5p	↓	Qatar	29 *	Peripheral blood
Pimenta et al. [65]	miR-200c-3p	↑	Brazil	72—39	Saliva
Sabbattinelli et al. [66]	miR-21-5p, 126-3p	↓,↓	Italy	29—29	Plasma
Saulle et al. [67]	miR-21-5p, 29a, 29c, 92a-3p, 150-5p, 155-5p	↑,↑,↑,↑,↑,↑	Italy	15—6	Plasma
Wilson et al. [68]	miR-29b-3p, 150-5p, 320e, 451a	↑,↑,↑,⇅	England	58 *	Plasma
Wu et al. [69]	miR-92b-3p, 92b-5p, 320b, 320c	↓,↑,↑,↑	USA	6—7	Nasopharyngeal Swab

* No control group, ↑ upregulation, ↓ downregulation, ⇅up and downregulation.

**Table 3 ijms-24-03574-t003:** Genes regulated by microRNAs.

Gene Symbol	*p*-Value	FDR	Odd Ratio	miRNAs
MCL1	2.22 × 10^−9^	8.54 × 10^−6^	0.071951	miR-29a-3p; miR-29b-3p; miR-29c-3p; miR-200a-3p; miR-92b-3p; miR-92a-3p; miR-320b; miR-320c; miR-320d
COL4A2	4.85 × 10^−9^	9.32 × 10^−6^	0.015698	miR-29b-3p; miR-29c-3p; miR-29a-3p; miR-155-5p; miR-92a-3p
MMP2	9.66 × 10^−7^	0.000161	0.040554	miR-29b-3p; miR-451a; miR-21-5p; miR-29c-3p; miR-29a-3p
SIRT1	2.42 × 10^−6^	0.0003	0.048403	miR-29c-3p; miR-126-3p; miR-155-5p; miR-200c-3p; miR-92a-3p
VEGFA	6.92 × 10^−6^	0.000682	0.117738	miR-126-3p; miR-29b-3p; miR-150-5p; miR-200c-3p; miR-29c-3p; miR-21-5p; miR-29a-3p
STAT3	0.000113	0.003962	0.104656	miR-155-5p; miR-21-5p; miR-92a-3p; miR-200a-3p; miR-29b-3p
MMP16	0.000542	0.369273	0.292207	miR-150-5p; miR-155-5p; miR-200a-3p; -200c-3p; miR-29a-3p, -29b-3p, -29c-3p; miR-320b, -320c, -320d; miR-92a-3p, -92b-3p
DICER1	0.001145	0.014206	0.122643	miR-29a-3p; miR-29c-3p; miR-21-5p; miR-200a-3p

FDR: False Discovery Rate; MCL1: Myeloid leukemia 1; COL4A2: Collagen, type IV, alpha 2; MMP2: matrix metallopeptidase 2; SIRT1: Sirtuin 1; VEGFA: Vascular endothelial growth factor A; STAT3: Signal transducer and activator of transcription 3; MMP16: matrix metallopeptidase 16; DICER1: dicer 1, ribonuclease III. Constructed with MIENTURNET [112].

## Data Availability

No new data were created or analyzed in this study. Data sharing is not applicable to this article.

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
