# Peer review of "Role of the MicroRNAs in the Pathogenic Mechanism of Painful Symptoms in Long COVID: Systematic Review"

_ijms, 2023, doi:10.3390/ijms24043574_

Round 1
Reviewer 1 Report
The current article by Samuel Reyes-Long and al., reviews the role of the microRNAs in the pathogenic mechanism of paint-ful symptoms in long COVID. The title of the paper is in line with the body of the manuscript. The topic is current and timely and very important for the world's scientific community and the authors have written a clear and detailed review and the material is well presented, although the authors do not express particular personal ideas on the future prospects for investigation. The references used are suitable and it is new and updated material, I believe that the article can be accept. I have below some observations that I think they can make the paper more complete:
1) Please check the manuscript for spelling and grammar mistakes
2) Line 38: “thar”
3) Lines 90-93: “OR and AND”
4) Line 113: “al”
5) Line 165: “ill”
6) Line 228: “nasopharynx”
7) Line 30: make this line better
8) Line 54: something seems to be missing after “by”…
9) The authors should include in their discussion and their conclusions greater personal opinion
Author Response
Reviewer 1.
The current article by Samuel Reyes-Long and al., reviews the role of the microRNAs in the pathogenic mechanism of paint-ful symptoms in long COVID. The title of the paper is in line with the body of the manuscript. The topic is current and timely and very important for the world's scientific community and the authors have written a clear and detailed review and the material is well presented, although the authors do not express particular personal ideas on the future prospects for investigation. The references used are suitable and it is new and updated material, I believe that the article can be accept. I have below some observations that I think they can make the paper more complete:
Thank you so much for your time and comments about our work, we appreciate the comments and we performed the changes you recommended.
1) Please check the manuscript for spelling and grammar mistakes
- All the manuscript was carefully checked.
2) Line 38: “thar”
- Corrected.
3) Lines 90-93: “OR and AND”
- Corrected.
4) Line 113: “al”
- Corrected.
5) Line 165: “ill”
- The term “critically ill” was changed.
6) Line 228: “nasopharynx”
- Changed to “nasopharyngeal”.
7) Line 30: make this line better
- Corrected.
8) Line 54: something seems to be missing after “by”...
- The word was deleted.
9) The authors should include in their discussion and their conclusions greater personal opinion
- A small statement was added in the Conclusion.

Reviewer 2 Report
This is an interesting and informative systematic review of the contribution of microRNAs to long COVID. The authors have identified 42 articles for inclusion and summarized the conclusions from these papers. Separate sections are dedicated to each miRNA/miRNA family of interest. The discussion provides two potential models for how these miRNAs may influence the symptoms oberved in long COVID patients. One model relates to effects on the IL-6/STAT3 pathway. A second model relates to effects on the blood-nerve barrier. These models appear reasonable and provide hypotheses for further testing. Overall, the review is organized and well written. The authors should be careful to clearly distinguish which sub-groups exihibit higher/lower miRNA expression when comparing experimental groups. Also, efforts to account for discrepancies between studies are encouraged where possible. One variable is disease severity and this factor is mentioned, it could be further emphasized. Finally, it would be helpful for the reader to provide a table correlating each miRNA with its mRNA/protein target where known.
Author Response
Reviewer 2.
This is an interesting and informative systematic review of the contribution of microRNAs to long COVID. The authors have identified 42 articles for inclusion and summarized the conclusions from these papers. Separate sections are dedicated to each miRNA/miRNA family of interest. The discussion provides two potential models for how these miRNAs may influence the symptoms oberved in long COVID patients. One model relates to effects on the IL-6/STAT3 pathway. A second model relates to effects on the blood-nerve barrier. These models appear reasonable and provide hypotheses for further testing. Overall, the review is organized and well written. The authors should be careful to clearly distinguish which sub-groups exihibit higher/lower miRNA expression when comparing experimental groups. Also, efforts to account for discrepancies between studies are encouraged where possible. One variable is disease severity and this factor is mentioned, it could be further emphasized. Finally, it would be helpful for the reader to provide a table correlating each miRNA with its mRNA/protein target where known.
We appreciate your time for reading and commenting our manuscript. A small table was added with genes and the miRNAs that regulate them.
